# Childhood Maltreatment and Its Interaction with Hypothalamic–Pituitary–Adrenal Axis Activity and the Remission Status of Major Depression: Effects on Functionality and Quality of Life

**DOI:** 10.3390/brainsci11040495

**Published:** 2021-04-13

**Authors:** Neus Salvat-Pujol, Javier Labad, Mikel Urretavizcaya, Aida De Arriba-Arnau, Cinto Segalàs, Eva Real, Alex Ferrer, José Manuel Crespo, Susana Jiménez-Murcia, Carles Soriano-Mas, José Manuel Menchón, Virginia Soria

**Affiliations:** 1Neurosciences Group—Psychiatry and Mental Health, Institut d’Investigació Biomèdica de Bellvitge (IDIBELL), Department of Psychiatry, Hospital Universitari de Bellvitge, 08907 Barcelona, Spain; nsalvat@bellvitgehospital.cat (N.S.-P.); murretavizcaya@bellvitgehospital.cat (M.U.); adearriba@bellvitgehospital.cat (A.D.A.-A.); csegalas@bellvitgehospital.cat (C.S.); ereal@bellvitgehospital.cat (E.R.); aferreralberti@gmail.com (A.F.); jmcrespo@bellvitgehospital.cat (J.M.C.); sjimenez@bellvitgehospital.cat (S.J.-M.); csoriano@idibell.cat (C.S.-M.); jmenchon@bellvitgehospital.cat (J.M.M.); 2Department of Mental Health, Corporació Sanitària Parc Taulí, 08208 Sabadell, Spain; 3Centro de Investigación Biomédica en Red de Salud Mental (CIBERSAM), Carlos III Health Institute, 28029 Madrid, Spain; jlabad@csdm.cat; 4Department of Clinical Sciences, School of Medicine, Universitat de Barcelona, 08907 Barcelona, Spain; 5Department of Mental Health, Consorci Sanitari del Maresme, Institut de Investigació i Innovació Parc Taulí (I3PT), 08304 Barcelona, Spain; 6Centro de Investigación Biomédica en Red de Fisiopatología Obesidad y Nutrición (CIBEROBN), Carlos III Health Institute, 28029 Madrid, Spain; 7Department of Psychobiology and Methodology of Health Sciences, Universitat Autònoma de Barcelona, 08193 Bellaterra, Spain

**Keywords:** major depressive disorder, childhood maltreatment, cortisol, functionality, quality of life

## Abstract

Relationships among childhood maltreatment (CM), hypothalamic-pituitary-adrenal (HPA) axis disturbances, major depressive disorder (MDD), poor functionality, and lower quality of life (QoL) in adulthood have been described. We aimed to study the roles of the remission status of depression and HPA axis function in the relationships between CM and functionality and QoL. Ninety-seven patients with MDD and 97 healthy controls were included. The cortisol awakening response, cortisol suppression ratio in the dexamethasone suppression test, and diurnal cortisol slope were assessed. Participants completed measures of psychopathology, CM, functionality, and QoL. Multiple linear regression analyses were performed to study the relationships between CM and functionality and QoL. Only non-remitted MDD patients showed lower functionality and QoL than controls, indicating that depressive symptoms may partly predict functionality and QoL. Cortisol measures did not differ between remitted and non-remitted patients. Although neither HPA axis measures nor depression remission status were consistently associated with functionality or QoL, these factors moderated the effects of CM on functionality and QoL. In conclusion, subtle neurobiological dysfunctions in stress-related systems could help to explain diminished functionality and QoL in individuals with CM and MDD and contribute to the persistence of these impairments even after the remission of depressive symptoms.

## 1. Introduction

Major depressive disorder (MDD) is a recurrent psychiatric condition estimated to affect more than 320 million people worldwide and has been identified as the leading cause of disability [1]. In fact, functional impairment is one of the core features of MDD [2]. As a chronic and recurrent illness, MDD is also associated with reduced quality of life (QoL) [3,4].

The decrease in functionality and reduced QoL generally occur in parallel with the worsening of depressive symptoms [5,6]. However, some authors have observed the persistence of substantial functional impairment and lower QoL in patients with subthreshold symptoms of depression or in remitted patients; thus, functional impairment and poor QoL are considered as enduring subsyndromal symptoms [4,7,8,9]. Impaired functioning and poor QoL may also predict the relapse of depressive episodes [4,10,11,12,13]. Therefore, the recovery of functionality and QoL is critical for depression patients to achieve and remain in remission, allowing them to return to productive and fulfilling daily lives [7,14].

Apart from disability and poor QoL, MDD has been linked to abnormalities in the hypothalamic-pituitary-adrenal (HPA) axis, which is a major physiological stress response system. Specifically, the following alterations have been described in MDD: hyperactivity of the HPA axis [15]; disruption of circadian HPA rhythms; impaired negative feedback responses [16], including altered feedback inhibition by endogenous glucocorticoids [17]; and increased cortisol awakening responses [18].

Nevertheless, not all patients with MDD present with HPA axis disturbances. Childhood maltreatment (CM) is thought to play a role in this variability, as it induces long-lasting neurobiological changes that resemble the neuroendocrine features of MDD [19], including HPA axis abnormalities [20,21]. Previous studies suggest that the severity and the type of childhood maltreatment might be linked to different disturbances in HPA axis function in adulthood. Nevertheless, until now, it is not clear whether this influence leads to an enhancement or a suppression of cortisol secretion [22,23,24,25,26,27,28,29]. It has been proposed that neuroendocrine abnormalities in MDD may be partly explained by the effects of CM on the HPA axis and might represent an individual’s susceptibility to developing depression in response to stress rather than being a consequence of depression [19]. Similarly, studies have linked CM with long-term HPA axis dysfunction independent of depression severity [25,30]. 

Lower functionality and QoL have been systematically documented in adults with a history of CM, even decades after the maltreatment ends [31,32,33,34,35], and there is a risk for disability or poorer QoL increase as a function of CM severity [36,37]. Different types of CM have also been linked to different degrees of impairment of QoL. For example, emotional neglect had the strongest influence on reduced QoL, followed by sexual and physical abuse [33,34]. Disability caused by mental health problems has been associated with childhood emotional abuse and physical neglect in one study [32], and with childhood physical abuse in another [38]. Meanwhile, poor physical functioning in adulthood was associated with neglect, psychological, and sexual abuse [31]. It is noteworthy that the results are mixed with regard to sexual abuse, as some studies strongly link it to disability [39], while others do not find an association between sexual abuse and disability caused by mental health problems [38]. The associations between CM and QoL or disability could be partially explained by depressive symptoms [34]. This is not surprising, since it has been reported that exposure to CM has long-lasting deleterious effects on mental health, including an increased susceptibility to developing depression in adulthood [40].

Instead, there is limited research assessing the effects of cortisol levels on functionality and QoL; most of the research indicates that higher cortisol levels are related with greater functional deficits [41] and poorer QoL [3,42] in samples of adults with depression and controls. In a study by Tang et al. [3], cortisol levels remained inversely related to the domain of psychological QoL after controlling for the severity of depression, thus providing novel evidence linking dysregulation of the HPA axis to QoL impairments in MDD independent of the severity of depressive symptoms.

Previous studies in children acknowledge the importance of the social environment that subjects grow up in when assessing the links among cortisol variables, psychopathology, and QoL [43]. To our knowledge, there is scarce information regarding the potential roles of HPA axis function and the remission status of MDD in the effects of CM on disability and QoL in adults.

CM affects functionality and QoL in adulthood, and all these variables seem to be associated with HPA axis dysfunction and depression. Thus, we hypothesized that the associations between different types of CM and both functionality and QoL are moderated by HPA axis function and MDD remission status. Our main aim was to investigate the effects of alleged CM on the functionality and QoL in adulthood and to assess the potential moderating effect of HPA axis function and MDD remission status on these relationships. Additionally, as a secondary aim in the present study, we sought to determine whether the known effects of HPA axis disturbances on functionality and QoL are moderated by the remission status of MDD.

## 2. Materials and Methods

### 2.1. Sample

The sample matches the one used in our previous work [44]. It included 97 patients with MDD (70.1% females, mean age 59.8 ± 11.7 years) recruited from the Psychiatry Department at Hospital Universitari de Bellvitge (Barcelona) and 97 healthy controls (HCs; 66.0% females, mean age 56.6 ± 11.9 years) recruited by advertisements from the same geographic area. Exclusion criteria were age less than 18 years, a diagnosis of other psychiatric disorders including substance abuse or dependence (except nicotine), intellectual disability, neurological disorders, severe medical conditions, pregnancy or puerperium, electroconvulsive therapy in the previous year, and corticosteroid treatment in the previous three months.

Although the cognitive assessment and HPA axis measures of the sample are the same as in the previous study [44], the current manuscript tested different hypotheses and included new data on functionality and QoL.

### 2.2. Clinical Assessment

Patients were assessed using the Mini-International Neuropsychiatric Interview (MINI) [45] and met the DSM-IV-TR criteria for MDD [2]. Depression severity was assessed with the 17-item Hamilton Depression Rating Scale (HDRS) [46]. Remitted depression was defined as an HDRS score <8.

HCs had no current or past history of psychiatric disorders as assessed with a semi-structured interview and a score below 7 on the 28-item Spanish adaptation of the Goldberg General Health Questionnaire (GHQ-28) [47].

Although some participants experienced CM, none met the DSM-IV-TR criteria for post-traumatic stress disorder [2].

Sociodemographic and clinical variables were assessed using a semi-structured interview. Both sleep quality (evaluated with the Pittsburgh Sleep Quality Index (PSQI) [48]) and body mass index (BMI) (calculated with the formula weight (kg)/height (m^2^)) were assessed as potential confounders in the analyses.

Exposure to CM was retrospectively assessed using the Childhood Trauma Questionnaire (CTQ) [49], which is a self-report inventory with 28 items rated on a five-point Likert scale to measure the frequency of events. The CTQ yields a total score and scores for five subscales corresponding to different types of maltreatment (emotional abuse, physical abuse, sexual abuse, emotional neglect, and physical neglect). Subscale scores range from 5 (no history of abuse or neglect) to 25 (history of extreme abuse or neglect). Cut-off scores for none, low, moderate, and severe exposure to maltreatment are provided for each subscale. Exposure to CM was defined as having at least one CTQ subscale score that was at or above the cut-off value for moderate exposure (emotional abuse ≥13; physical abuse ≥10; sexual abuse ≥8; emotional neglect ≥15; and physical neglect ≥10) [49]. Individuals classified as negative on all subscales were considered to have no exposure to CM.

To control for anxiety, as a correlate of personality trait and current stressors, we used the State-Trait Anxiety Inventory (STAI) [50]. The STAI is a self-report inventory that assesses two independent concepts of anxiety: anxiety as a state (STAI-state), which is a transient emotional condition, and anxiety as a trait (STAI-trait), which is a relatively stable propensity for anxiety. The STAI consists of 40 items, including 20 items for each concept. For the STAI-state subscale, items are rated from 0 to 3 depending on their intensity; for the STAI-trait subscale, items are rated from 0 to 3 depending on their frequency. Total scores range from 0 to 60 for each concept.

The Social Adaptation Self-evaluation Scale (SASS) [51,52] was used to evaluate functional impairment. It consists of 21 items with four response options (from 0 to 3) to assess functionality in different areas, including work, family, leisure, social relationships, and motivation/interests. The total score, obtained by adding scores on each individual item, ranges between 0 and 60. The proposed cut-off points are as follows: <25, social impairment; 25–52, normal; and >52, pathological over-adaptation [51].

The 3-level version of the Euro Quality of Life 5-dimensional questionnaire (EQ-5D-3L) [53] was used to evaluate self-perceived health-related QoL. It consists of three parts:(a)EQ-5D-3L offers descriptive information on five dimensions (mobility, self-care, daily activities, pain/discomfort, and anxiety/depression) with three response options (1 = “no problem”, 2 = “some problems”, 3 = “serious problems”). The results can be presented as health profiles (i.e., 11111 indicates no problems in any of the levels).(b)These health profiles can be converted into a single summary index (EQ-5D-index) by applying a formula that attaches weights to each of the levels in each dimension. The index is calculated by deducting the appropriate weights from 1, which is the value for full health (i.e., state 11111).(c)EQ-5D-VAS: This Visual Analogue Scale or “health thermometer” offers quantitative information for health self-assessments. It ranges from 0 (the worst health one can imagine) to 100 (the best health one can imagine).

### 2.3. Salivary Cortisol Measurements

Participants collected saliva samples at home for cortisol analyses shortly after clinical assessments using Salivette^®^ (Sarstedt AG & Co., Nümbrecht, Germany) containers and following the same process as in our previous studies [44,54]. Participants were instructed to collect all samples during a regular day, avoiding stressful situations and intense physical activity. Eating, drinking, smoking, or brushing teeth were not allowed in the 15 min prior to the collection of each sample. Samples were obtained at awakening (T1), 30 min (T2) and 60 min (T3) after awakening, at 10 a.m. (T4) and at 11 p.m. (T5) on the same day. Participants were requested to take 0.25 mg dexamethasone just after T5, and a sample was obtained at 10 a.m. the following day (T6). Samples were stored in refrigerators and returned personally by each participant within one week from collection. The same day that the samples were received, they were stored at −20 °C and later sent to the BioBanc from the Institut d’Investigació Sanitària Pere Virgili (IISPV) for centrifugation (3000 rpm for 5 min) and aliquotation, after which they were frozen at −20 °C until analysis by enzyme-linked immunosorbent assay (IBL International, Hamburg, Germany) to determine saliva cortisol levels.

HPA axis function was assessed using three dynamic tests:Cortisol awakening response (CAR): CAR is a physiological response to awakening that consists of a rise in cortisol levels following morning awakening [55]. The CAR was calculated using the area under the curve with respect to the increase in cortisol [56], including T1, T2, and T3 samples. The CAR was assessed on one day only; as dexamethasone was administered at 11 p.m., we dismissed the possibility of collecting further CAR sampling the next day.The cortisol suppression ratio in the dexamethasone suppression test (DSTR) using a very low dose of dexamethasone (0.25 mg) (see [44] for more information on this decision): This ratio provides information about the feedback inhibition of the HPA axis. Intake of dexamethasone, a synthetic glucocorticoid receptor agonist, results in the suppression of the secretion of cortisol by the adrenal gland. The DSTR is defined as the ratio of cortisol T4/cortisol T6. Higher DSTRs indicate greater suppression of cortisol secretion after dexamethasone administration, and a lack of suppression indicates reduced feedback sensitivity and is considered a measure of glucocorticoid resistance [16,17].Diurnal cortisol slope: This slope represents the rate of decline in cortisol levels across the day from morning to evening and it reflects the diurnal cortisol rhythm. It was calculated using T4 and T5 samples. Steeper cortisol slopes reflect more preserved diurnal cortisol rhythms [23].

### 2.4. Statistical Analyses

Data processing was performed using SPSS 21 (SPSS, IBM Corp., Armonk, NY, USA). We normalized the data as described in previous works [44,54]. As explained elsewhere [44], outliers were defined as transformed cortisol values that were located more than three standard deviations from the mean, and they were excluded from analyses. The number of outliers were 2 samples for the HC group and 7 samples for the non-remitted MDD group, without outliers in the remitted MDD group.

Categorical data were compared across groups (HCs vs. remitted MDD vs. non-remitted MDD) with Chi-square tests. The independent samples t-test and ANOVA were applied to compare continuous variables between two or three groups, respectively. Bonferroni correction was used for post hoc analyses in ANOVA.

Exploratory partial correlation analyses adjusted for gender, age, and years of education were used to explore the relationships between cortisol, CM, functionality, and QoL measures. We conducted a stratified analysis by diagnosis (HCs vs. remitted MDD vs. non-remitted MDD). The statistical significance level was set at *p* < 0.05 (two-tailed). As these analyses were exploratory in nature, and the main hypotheses of our study were tested with other multivariate analyses, we did not correct for multiple comparisons [57].

The hypotheses testing the relationship between types of CM and functionality (SASS) or QoL, while controlling for covariates and the potential moderating effects of HPA axis measures and/or remission status were tested with separate multiple linear regression (MLR) analyses. In these analyses, which included all participants, the SASS, EQ-5D-index, and EQ-5D-VAS were considered the dependent variables. WE created two dummy variables to define MDD clinical status (remission and non-remission) with respect to HCs, which was considered the reference category. Both variables were used as independent variables in the equation. All independent variables were entered into the equation. Interaction terms between CM measures, cortisol variables, and remission status were tested in a final step. We controlled for potential confounders, including gender, age, years of education, BMI, tobacco consumption, sleep quality, anxiety (STAI-state and STAI-trait), and cortisol at awakening (T1) [58] in all analyses. Only significant interaction terms were kept in the final equation. We included sequential MLR steps (models) with a hierarchical approach in order to verify changes in R^2^ (proportion of explained variance in the dependent variable in the model) when adding the selected variables to the equation in consecutive steps. The statistical significance level was set at *p* < 0.05 (two-tailed).

In the previous MLR analyses, both MDD groups (remitted and non-remitted) were compared to HCs. We repeated all MLR equations without including HCs to compare both MDD groups (remitted vs. non-remitted) and explore the specific moderating effect of MDD remission status. Non-remission was considered the reference category.

## 3. Results

### 3.1. Univariate Analyses

The demographic, clinical and cortisol variables for the three study groups are described in detail elsewhere [44] and in Table 1 from the Appendix A. In summary, HCs were significantly younger than patients with remitted MDD, and there were no differences in gender between groups. HCs had undergone more education years than both MDD groups. BMI was significantly higher in non-remitted MDD compared with HCs. Finally, patients with non-remitted MDD presented with poorer sleep quality than the other study groups. We found no significant differences between groups regarding tobacco and alcohol consumption. There were no statistically significant differences in the clinical variables of depression (age of onset; percentage of melancholic and atypical symptoms; number of depressive episodes and hospitalizations; suicide attempts) between remitted and non-remitted patients with MDD except for HDRS scores, which, as expected, were significantly higher in non-remitted patients. STAI (state and trait) scores were also significantly higher in non-remitted patients. HPA axis measures did not differ between groups except for evening cortisol levels, which was higher in non-remitted MDD patients than in HCs.

Table 1 contains information on functionality, QoL, and CM variables. Patients with non-remitted MDD reported poorer functionality and lower QoL in all EQ-5D dimensions, the EQ-5D-index and the EQ-5D-VAS than HCs or remitted MDD patients. Remitted MDD patients reported more severe pain/discomfort and anxiety/depression problems and a lower EQ-5D-index than HCs. Patients with non-remitted MDD reported higher CTQ total scores than HCs and higher scores on the emotional abuse subscale than HCs and remitted MDD patients.

### 3.2. Partial Correlation Analyses

Partial correlation analyses are displayed in Appendix A. In patients with remitted MDD, the EQ-5D-VAS score was negatively correlated with emotional abuse scores (*r* = −0.580, *p* = 0.003) and CTQ total scores (*r* = −0.407, *p* = 0.048), while the diurnal cortisol slope was positively correlated with the EQ-5D-index (*r* = 0.457, *p* = 0.025) and EQ-5D-VAS scores (*r* = 0.433, *p* = 0.034). In patients with non-remitted MDD, emotional neglect scores were negatively correlated with the CAR (*r* = −0.398, *p* = 0.030). No other correlations between CTQ, cortisol, functionality, or QoL variables were found in either the MDD or HC groups.

### 3.3. Multiple Linear Regression Analyses in All Participants

#### 3.3.1. Functional Impairment

Table 2 displays the results of the MLR analysis, with SASS scores as the dependent variable. In the final model, STAI state and trait scores were associated with poorer functioning.

CM, depression remission status, and cortisol variables did not have an effect on functionality. However, HPA axis measures interacted with CM and depression remission status. More specifically, individuals with higher physical neglect scores and higher CARs presented with lower functionality, while sexual abuse was associated with lower functionality in participants with blunted CAR. The DSTR interacted with MDD remission status such that patients with remitted MDD and higher cortisol suppression after dexamethasone intake had more preserved functionality.

#### 3.3.2. Quality of Life

Table 3 and Table 4 display the results of the MLR analyses in all participants, with QoL as assessed by the EQ-5D-index and EQ-5D-VAS scores as the dependent variables.

Higher STAI state and trait scores and MDD non-remission status were associated with lower QoL as assessed with the EQ-5D-index scores. CM and HPA axis function per se were not associated with EQ-5D-index scores, but there was an interaction between the DSTR and MDD non-remission status, suggesting that patients with non-remitted depression who had a higher DSTR presented with higher QoL. Another significant interaction was found between the remission status of MDD and CTQ scores, indicating poorer QoL in remitted patients with higher physical neglect scores.

The STAI state score had a negative effect on QoL as assessed with the visual analogue scale. The STAI trait score and CM were not associated with the EQ-VAS score. MDD remission status was associated with higher EQ-VAS scores, but this relationship was moderated by diurnal cortisol slope and emotional abuse scores. This means that remitted MDD patients with steeper diurnal cortisol slopes or higher emotional abuse scores presented with poorer QoL. A more preserved CAR was also associated with higher EQ-VAS scores, but emotional neglect had a negative moderation effect on this association. An additional interaction was found between non-remitted MDD and sexual abuse, suggesting that patients with non-remitted MDD and higher scores on sexual abuse presented poorer QoL (measured by the EQ-VAS).

### 3.4. Multiple Linear Regression Analyses in MDD Patients Only

We repeated all MLR analyses in MDD patients only (see Appendix A).

#### 3.4.1. Functional Impairment

Physical abuse (*β* = −0.372, *p* = 0.024) was negatively associated with functional impairment in MDD patients, while emotional neglect (*β* = 0.636, *p* = 0.013) was positively associated with functional impairment. STAI state (*β* = −0.312, *p* = 0.039) and STAI trait (*β* = −0.462, *p* = 0.015) scores exerted negative effects on SASS scores.

Steeper diurnal cortisol slopes were associated with better functioning (*β* = −10.135, *p* = 0.011). Nevertheless, an interaction with emotional neglect was found, indicating that patients with emotional neglect and a steeper diurnal cortisol slope presented with lowered functioning (*β* = 10.462, *p* = 0.004).

On the other hand, the interaction between the DSTR and MDD remission status indicates that patients with remitted MDD and a more preserved HPA axis negative feedback response present with better functioning (*β* = 0.232, *p* = 0.028).

#### 3.4.2. Quality of Life

CM was not associated with QoL as assessed by EQ-5D-index scores, but STAI-trait scores were associated with EQ-5D-index scores (*β* = −0.471, *p* = 0.016), suggesting that individuals with higher levels of trait anxiety present with lower QoL. Remission status of depression and HPA axis measures were not associated with EQ-5D-index scores, but we found a significant interaction between the CAR and MDD remission status (*β* = 0.356, *p* = 0.039), indicating that patients with remitted depression and a higher CAR would present a higher QoL.

STAI-state (*β* = −0.365, *p* = 0.008) scores and steeper diurnal cortisol slopes (*β* = 0.244, *p* = 0.014) had a negative effect on EQ-5D-VAS scores. Meanwhile, other HPA axis measures, STAI-trait scores, CM and MDD remission status were not associated with EQ-5D-VAS scores, and we did not find any significant interactions between variables.

## 4. Discussion

To our knowledge, this is the first study to report the moderating effects of HPA axis function and MDD remission status on the associations between CM and functionality and QoL.

Total CTQ scores were higher in non-remitted MDD patients than in HCs. Our results are in accordance with previous studies that have found a link between depression symptoms and higher levels of CM [59]. Despite these findings, we did not find any significant differences between groups in any of the CTQ subscale scores except for emotional abuse.

In univariate analyses, we found that remitted MDD patients and HCs seemed to differ from non-remitted depressed patients in terms of functionality and QoL. This finding is consistent with previous studies reporting that depressive symptoms (or their persistence) may partly predict functionality and QoL [5,13,60,61].

When exploring the relationship between childhood trauma and outcome variables (functionality, QoL), only physical abuse had a negative impact on functionality in the analysis performed using the MDD sample, contrasting with previous reports of associations between different types of CM and disability or QoL [31,32,33,34,38]. These differences may be due to methodological issues, as previous studies did not take into account HPA axis function or depression remission status. In fact, in our study, we found that HPA axis measures and remission status of depression moderated the effects of childhood trauma on functionality and QoL.

### 4.1. Predictors of Functioning

The CAR moderated the association between childhood trauma dimensions and functionality, with a distinct pattern in participants who experienced sexual abuse (a blunted CAR, which represents an abnormal secretion pattern of cortisol after awakening, was associated with poorer functionality) and physical neglect (a blunted CAR was associated with better functionality). These findings suggest that abuse and neglect show a distinct pattern in relation to the effects of the CAR on functionality.

Previous studies considering different subtypes of childhood maltreatment and CAR have found that only sexual abuse was associated with an increased CAR [62]. In our study, an increased CAR, which represents the physiological secretion pattern of cortisol after awakening, was associated with better functionality in participants who experienced sexual abuse. As previous studies suggest that a blunted CAR is associated with hippocampal damage [63], it is also possible that the moderating effects of some HPA axis indices (e.g., the CAR) on functionality measures could also involve cognitive domains that are mediated by the hippocampus. In fact, CM has been linked to smaller volumes or dysfunction of the hippocampus and poorer performance in memory and on executive functioning tasks [64,65,66,67].

There is not a clear explanation for the inverse associations between the CAR and functionality in neglected participants. There is a debate about the potential different neurobiological consequences of abuse and neglect [65], although there is little information regarding people with MDD and functionality-related outcomes. It is important to conduct further studies to replicate our results and to determine whether subtle alterations in the CAR may contribute to the persistence of functional impairments in individuals who experience physical neglect or sexual abuse.

Some results regarding the diurnal cortisol pattern were also found. Steeper cortisol slopes, which are considered to indicate healthier profiles than flattened cortisol slopes [23], were associated with higher functionality in the MDD sample, which is consistent with previous literature linking flatter cortisol slopes with poorer functionality in the context of MDD [41]. Our study suggests that the cortisol diurnal slope moderated the association between emotional neglect and functionality in patients with MDD.

The DSTR moderated the association between remission status and functionality. Cortisol non-suppression in response to dexamethasone, which suggests a lower HPA axis negative feedback response, has been reported in 40–60% of depressed patients [68]. Disturbances in the HPA axis, particularly hypercortisolemia, have been linked to poorer functionality in patients with depression [41]. In our study, a more stable HPA axis negative feedback response, as defined by a greater DSTR, was associated with improved functionality in remitted MDD patients. Our results suggest that an HPA axis measure of GR resistance (a low DSTR) could be a trait marker of functional impairment, because this association was found in patients with remitted MDD. Nevertheless, the effect of the interaction between MDD remission status and the DSTR on functionality was maintained in the analyses conducted in MDD patients only, suggesting that alterations in the feedback cortisol response might be state markers of functional impairment in remitted MDD patients.

### 4.2. Predictors of QoL

HPA axis measures and depression remission status also moderated the effects of childhood trauma on QoL.

People who experienced emotional neglect and had a blunted CAR showed improved QoL (measured by the EQ-VAS). As in functionality, there is not a clear explanation for the inverse associations between the CAR and QoL in neglected participants [65], so further studies are needed to replicate these results and to determine whether subtle alterations in the CAR may contribute to the persistence of poor QoL in individuals with a history of emotional neglect.

In the sub-analysis including only MDD patients, an interaction between remission and the CAR was found for the QoL index (an increased CAR was associated with better QoL). Our findings suggest that subtle alterations in this HPA axis measure may contribute to the persistence of QoL disturbances in patients with remitted depression.

The diurnal cortisol slope also moderated the association between MDD remission status and QoL (measured by the EQ-5D-VAS). In the sample containing only patients with MDD, flatter diurnal cortisol slopes were found to be associated with higher QoL, while the interactions observed in the whole sample (between remission status and the diurnal cortisol slope) could not be replicated.

A more stable HPA axis negative feedback response, as defined by a greater DSTR, was associated with better QoL (measured by the EQ-5D-index) in non-remitted MDD patients, but only in the analysis including all participants, suggesting that alterations in the feedback cortisol response might be trait markers of poor QoL in MDD.

### 4.3. Limitations and Methodological Issues

Some limitations and methodological issues need to be recognized. All patients in our sample received antidepressant treatment according to their clinical needs. Although pharmacological treatment may have influenced cortisol measures, there were no significant differences in antidepressant type or dose between the two MDD groups (data not shown) (see Salvat-Pujol et al., 2017). Patients were recruited from a tertiary source, which may differ from community-based patients, thereby limiting the generalization of these results.

Cortisol levels were measured only once. Although some previous studies collected multiple cortisol samples after dexamethasone administration [69], others only included one post-dexamethasone sample [18]. In fact, the DST has shown relatively good individual stability over time [70,71]. Even so, measuring the CAR using one sampling day might be considered a limitation according to recent expert consensus guidelines that recommend obtaining CAR data over two or more sampling days [58]. Nevertheless, our sampling procedure complies with that described in the Netherlands Study of Depression and Anxiety (NESDA) [72]. Both the NESDA and our study were designed before the publication of the CAR guidelines.

Although some studies have indicated that responses on some tests could be influenced by patients’ psychological states and their insight into the psychiatric disorder or that recall bias could affect the results, the CTQ [67], PSQI [48], SASS [52], and EQ-5D [73] have proven to be valid instruments with good internal consistency and suitable for clinical and research settings. Finally, the cross-sectional design precludes causal inferences, and longitudinal studies are needed to address this issue.

## 5. Conclusions

Our study provides novel evidence linking subtle alterations in the HPA axis to the persistence of functional impairment and diminished QoL in individuals with CM and MDD, even after the remission of depressive symptoms. These findings shed light on previous literature assessing functionality- and QoL-related changes in relation to the improvement or worsening of affective symptoms. Thus, our findings highlight the roles of both exposure to CM and HPA axis functionality in achieving full functional recovery and perceived well-being in patients with MDD.

Our results lend further support to the need for a more exhaustive assessment of CM exposure in patients with MDD. A better understanding of how exposure to different types of CM affects functionality and QoL in adulthood will help to develop strategies that facilitate the implementation of early intervention and prevention programs to reduce or prevent CM deleterious effects on functionality and QoL in the long-term.

In view of our results, future studies should determine whether neuroendocrine measures may be useful in monitoring the risk of functional decline and loss of QoL in individuals with CM and depression, even in remitted episodes.

## Figures and Tables

**Table 1 brainsci-11-00495-t001:** Disability, quality of life, and childhood maltreatment measures by study groups.

	HC		Remitted MDD		Non Remitted MDD		Statistics
	*n* = 97		*n* = 44		*n* = 53		(χ^2^/T-test)
Disability measures							
SASS	42.80 (7.98)		40.95 (6.42)		31.63 (8.14)		F(178) = 34.106, *p* < 0.001 ^b,c^
Quality of life measures							
EQ-5D dimensions							
Mobility problems	1.15 (0.36)		1.20 (0.41)		1.54 (0.58)		F(176) = 13.283, *p* = 0.001 ^b,c^
Self-care problems	1.02 (0.15)		1.03 (0.16)		1.33 (0.52)		F(176) = 18.763, *p* = 0.001 ^b,c^
Daily activities problems	1.07 (0.25)		1.23 (0.42)		2.02 (0.67)		F(176) = 76.003, *p* < 0.001 ^b,c^
Pain/discomfort problems	1.33 (0.47)		1.63 (0.54)		1.98 (0.73)		F(176) = 20.951, *p* < 0.001 ^a,b,c^
Anxiety/depression problems	1.12 (0.36)		1.45 (0.55)		2.59 (0.58)		F(176) = 149.792, *p* < 0.001 ^a,b,c^
EQ-5D-Index	0.89 (0.14)		0.79 (0.16)		0.43 (0.22)		F(176) = 117.119, *p* < 0.001 ^a,b,c^
EQ-5D-VAS	84.00 (11.81)		76.22 (19.91)		40.77 (21.10)		F(168) = 104.029, *p* < 0.001 ^b,c^
Childhood Trauma Questionnaire	*n* (%)	mean (SD)	*n* (%)	mean (SD)	*n* (%)	mean (SD)	
CTQ—emotional abuse	7 (7.2)	7.08 (2.93)	5 (11.4)	7.23 (4.15)	10 (18.9)	9.26 (4.68)	χ^2^ = 5.169, *p* = 0.075; F(176) = 5.607, *p* = 0.004 ^b,c^
CTQ—physical abuse	4 (4.1)	5.88 (1.66)	5 (11.4)	6.02 (2.14)	5 (9.4)	6.65 (3.95)	χ^2^ = 2.960, *p* = 0.228; F(178) = 1.431, *p* = 0.242
CTQ—sexual abuse	8 (8.2)	5.55 (1.23)	5 (11.4)	5.65 (1.75)	6 (11.3)	6.04 (3.29)	χ^2^ = 0.569, *p* = 0.752; F(176) = −0.867, *p* = 0.418
CTQ—emotional neglect	8 (8.2)	9.14 (4.04)	5 (11.4)	9.75 (4.24)	8 (15.1)	10.94 (4.02)	χ^2^ = 1.844, *p* = 0.398; F(177) = 3.025, *p* = 0.051
CTQ—physical neglect	8 (8.2)	6.39 (2.20)	4 (9.1)	6.93 (2.25)	8 (15.1)	6.94 (2.91)	χ^2^ = 2.016, *p* = 0.365; F(178) = 1.127, *p* = 0.326
CTQ—total score	NA	33.96 (8.52)	NA	34.68 (9.51)	NA	40 (14.84)	F(173) = 5.074, *p* = 0.007 ^b^
Exposed to childhood maltreatment	22 (22.7)	NA	13 (29.5)	NA	15 (28.3)	NA	χ^2^ = 1.097, *p* = 0.578

Abbreviations: HC, healthy controls; MDD, major depressive disorder; SASS, Social Adaptation Self-evaluation Scale; EQ-5D, Euro Quality of Life 5-dimensions questionnaire; VAS, Visual Analogue Scale; CTQ, Childhood Trauma Questionnaire. All variables presented in mean (SD). ^a^ Significant ANOVA post hoc analyses (comparison between groups) with Bonferroni correction: HC vs. remitted MDD. ^b^ Significant ANOVA post hoc analyses (comparison between groups) with Bonferroni correction: HC vs. non-remitted MDD. ^c^ Significant ANOVA post hoc analyses (comparison between groups) with Bonferroni correction: remitted MDD vs. non-remitted MDD.

**Table 2 brainsci-11-00495-t002:** Results of multiple linear regression analyses exploring the association of childhood maltreatment, hypothalamic–pituitary–adrenal (HPA) axis function, and MDD remission status with disability (SASS) in all participants (*n* = 194).

	Model 1		Model 2		Model 3		Model 4		Final Model	
	R^2^ = 0.058		R^2^ = 0.392		R^2^ = 0.469		R^2^ = 0.491		R^2^ = 0.560	
	*β*	*p*	*β*	*p*	*β*	*p*	*β*	*p*	*β*	*p*
CTQ—emotional abuse	−0.091	0.462	0.030	0.770	−0.007	0.945	−0.045	0.676	−0.043	0.677
CTQ—physical abuse	−0.054	0.613	−0.109	0.213	−0.100	0.264	−0.069	0.453	−0.083	0.338
CTQ—sexual abuse	0.076	0.401	0.034	0.641	0.057	0.451	0.089	0.247	0.074	0.311
CTQ—emotional neglect	−0.207	0.060	−0.038	0.680	0.019	0.830	0.021	0.822	0.067	0.441
CTQ—physical neglect	0.088	0.412	0.010	0.905	0.014	0.869	0.000	0.998	−0.025	0.765
STAI-state			−0.411	<0.001	−0.278	0.022	−0.254	0.037	−0.233	0.046
STAI-trait			−0.231	0.047	−0.227	0.058	−0.226	0.068	−0.265	0.027
Gender					0.108	0.156	0.122	0.110	0.096	0.187
Age					0.025	0.757	0.028	0.741	0.062	0.441
Years of education					−0.011	0.891	−0.036	0.672	−0.021	0.793
BMI					−0.216	0.005	−0.221	0.005	−0.251	0.001
Tobacco consumption (cig/day)					−0.007	0.929	0.013	0.861	−0.052	0.466
Non-remitted MDD					−0.143	0.191	−0.146	0.185	−0.164	0.119
Remitted MDD					0.011	0.887	−0.002	0.984	−0.007	0.919
PSQI					−0.009	0.922	−0.041	0.668	0.013	0.889
Waking cortisol							−0.162	0.062	−0.155	0.061
CAR							−0.081	0.366	0.007	0.981
DSTR							0.064	0.393	0.091	0.203
Diurnal cortisol slope							0.058	0.466	0.025	0.740
CAR × CTQ—physical neglect									−0.670	0.003
CAR × CTQ—sexual abuse									0.563	0.015
DSTR × Remitted MDD									0.137	0.041

SASS score was considered as the dependent variable. *β*: standardized beta coefficient. Abbreviations: MDD, major depressive disorder; SASS, Social Adaptation Self-evaluation Scale; CTQ, Childhood Trauma Questionnaire; STAI: State-Trait Anxiety Inventory; BMI, body mass index; PSQI, Pittsburgh Sleep Quality Index; CAR, cortisol awakening response calculated to the increase in cortisol; DSTR, dexamethasone suppression test ratio. Analyses performed using transformed cortisol variables, outliers excluded. Non-significant interaction terms were excluded in the final equation.

**Table 3 brainsci-11-00495-t003:** Results of multiple linear regression analyses exploring the association of childhood maltreatment, HPA axis function, and MDD remission status with quality of life (EQ-5D-Index scores) in all participants (*n* = 194).

	Model 1		Model 2		Model 3		Model 4		Final Model	
	R^2^ = 0.079		R^2^ = 0.597		R^2^ = 0.683		R^2^ = 0.694		R^2^ = 0.719	
	*β*	*p*	*β*	*p*	*β*	*p*	*β*	*p*	*β*	*p*
CTQ—emotional abuse	−0.169	0.171	−0.006	0.945	−0.054	0.512	−0.034	0.689	−0.037	0.663
CTQ—physical abuse	−0.011	0.919	−0.090	0.211	−0.047	0.504	−0.057	0.429	−0.049	0.476
CTQ—sexual abuse	0.009	0.919	−0.046	0.446	−0.031	0.594	−0.047	0.434	−0.039	0.503
CTQ—emotional neglect	−0.194	0.076	0.026	0.733	0.034	0.628	0.027	0.700	0.016	0.822
CTQ—physical neglect	0.066	0.536	−0.029	0.689	0.021	0.759	0.016	0.816	0.067	0.352
STAI-state			−0.419	<0.001	−0.223	0.019	−0.242	0.011	−0.206	0.028
STAI-trait			−0.390	<0.001	−0.293	0.002	−0.274	0.005	−0.317	0.001
Gender					0.018	0.765	0.010	0.864	0.036	0.537
Age					−0.144	0.026	−0.156	0.019	−0.151	0.021
Years of education					0.023	0.725	0.012	0.854	0.009	0.881
BMI					0.019	0.749	0.042	0.488	0.066	0.266
Tobacco consumption (cig/day)					−0.060	0.291	−0.076	0.187	−0.083	0.141
Non-remitted MDD					−0.331	<0.001	−0.333	<0.001	−0.427	<0.001
Remitted MDD					−0.080	0.177	−0.074	0.224	0.375	0.102
PSQI					−0.069	0.349	−0.065	0.382	−0.044	0.546
Waking cortisol							0.108	0.112	0.106	0.108
CAR							0.091	0.195	0.090	0.188
DSTR							0.036	0.541	0.050	0.382
Diurnal cortisol slope							0.089	0.156	0.093	0.125
DSTR × Non-remitted MDD									0.141	0.023
Remitted MDD × CTQ—physical neglect									−0.469	0.044

EQ-5D-Index score was considered as the dependent variable. *β*: standardized beta coefficient. Abbreviations: MDD, major depressive disorder; EQ-5D, Euro Quality of Life 5-dimensions questionnaire; CTQ, Childhood Trauma Questionnaire; STAI, State-Trait Anxiety Inventory; BMI, body mass index; PSQI, Pittsburgh Sleep Quality Index; CAR, cortisol awakening response calculated to the increase in cortisol; DSTR, dexamethasone suppression test ratio. Analyses performed using transformed cortisol variables, outliers excluded. Non-significant interaction terms were excluded in the final equation.

**Table 4 brainsci-11-00495-t004:** Results of multiple linear regression analyses exploring the association of childhood maltreatment, HPA axis function, and MDD remission status with quality of life (EQ-VAS) in all participants (*n* = 194).

	Model 1		Model 2		Model 3		Model 4		Final Model	
	R^2^ = 0.073		R^2^ = 0.599		R^2^ = 0.697		R^2^ = 0.721		R^2^ = 0.773	
	*β*	*p*	*β*	*p*	*β*	*p*	*β*	*p*	*β*	*p*
CTQ—emotional abuse	−0.218	0.087	−0.066	0.443	−0.029	0.735	−0.030	0.721	0.089	0.288
CTQ—physical abuse	0.117	0.284	0.047	0.521	0.073	0.307	0.085	0.235	0.042	0.525
CTQ—sexual abuse	0.015	0.869	−0.035	0.574	−0.009	0.873	−0.009	0.875	0.084	0.217
CTQ—emotional neglect	−0.220	0.050	−0.010	0.893	0.013	0.849	−0.011	0.870	0.056	0.395
CTQ—physical neglect	0.148	0.177	0.055	0.448	0.047	0.492	0.032	0.630	−0.045	0.486
STAI-state			−0.488	<0.001	−0.249	0.009	−0.274	0.004	−0.253	0.004
STAI-trait			−0.320	0.001	−0.216	0.020	−0.159	0.092	−0.151	0.087
Gender					0.054	0.352	0.048	0.404	0.050	0.343
Age					−0.024	0.709	−0.060	0.358	−0.017	0.783
Years of education					−0.100	0.118	−0.134	0.036	−0.125	0.039
BMI					−0.070	0.238	−0.039	0.510	−0.076	0.168
Tobacco consumption (cig/day)					−0.026	0.652	−0.038	0.504	−0.075	0.168
Non-remitted MDD					−0.361	<0.001	−0.370	<0.001	−0.110	0.523
Remitted MDD					−0.012	0.842	−0.012	0.839	0.500	0.001
PSQI					−0.148	0.046	−0.178	0.016	−0.150	0.030
Waking cortisol							0.060	0.353	0.070	0.245
CAR							0.019	0.783	0.453	0.002
DSTR							0.065	0.262	0.062	0.244
Diurnal cortisol slope							0.176	0.004	0.057	0.376
Diurnal cortisol slope × Remitted MDD									0.211	0.005
CAR × CTQ—emotional neglect									−0.479	0.001
Remitted MDD × CTQ—emotional abuse									−0.412	0.002
Non-Remitted MDD × CTQ—sexual abuse									−0.362	0.041

EQ-5D-Index score was considered as the dependent variable. *β*: standardized beta coefficient. Abbreviations: MDD, major depressive disorder; EQ-5D, Euro Quality of Life 5-dimensions questionnaire; VAS: visual analogue scale; CTQ, Childhood Trauma Questionnaire; STAI, State-Trait Anxiety Inventory; BMI, body mass index; PSQI, Pittsburgh Sleep Quality Index; CAR, cortisol awakening response calculated to the increase in cortisol; DSTR, dexamethasone suppression test ratio. Analyses performed using transformed cortisol variables, outliers excluded. Non-significant interaction terms were excluded in the final equation.

## Data Availability

The datasets generated during the current study are not publicly available because of conditions stipulated by the consent form. However, additional data or analyses are available from the corresponding author upon reasonable request.

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
