# Peer review of "Childhood Maltreatment and Its Interaction with Hypothalamic–Pituitary–Adrenal Axis Activity and the Remission Status of Major Depression: Effects on Functionality and Quality of Life"

_brainsci, 2021, doi:10.3390/brainsci11040495_

Round 1

Reviewer 1 Report

Salvat-Pujol et al. have studied the correlation between childhood maltreatment (CM), saliva cortisol levels, and functionality and quality of life (QoL) in patients with major depression (MDD). They have found that non-remitted, but not remitted, MDD patients showed lower functionality and QoL than controls and no difference in the cortisol level between remitted and non-remitted patients of MDD. The authors concluded that “mild neurobiological dysfunctions in stress-related systems could help to explain diminished functionality and QoL in individuals with CM and MDD and contribute to the persistence of these impairments even after the remission of depressive symptoms”. The data are original and novel. I have a few minor concerns:

  1. What is the use of presenting Models 1 to 4, in addition to the final model? I may have missed it but have not appreciated the benefit of presenting Models 1 to 4.
  2. Outliers have been excluded in Tables 2, 3 and 4. How are the outliers defined and how many outliers have been excluded? I think such information could be helpful to provide confidence in the solidity of the data.
  3. Line 180 to 182, “storage Samples were stored in refrigerators and returned personally by each participant. After the samples were received, they were stored at -20 °C and later sent to the BioBanc…” How soon did the participants return the samples once collecting them? How soon did the investigators send the samples out to the BioBanc once they receive them from participants? Variations of the storage durations of samples may affect the level of cortisol, so clarifying the time duration for these storage time would be useful.

Author Response

Salvat-Pujol et al. have studied the correlation between childhood maltreatment (CM), saliva cortisol levels, and functionality and quality of life (QoL) in patients with major depression (MDD). They have found that non-remitted, but not remitted, MDD patients showed lower functionality and QoL than controls and no difference in the cortisol level between remitted and non-remitted patients of MDD. The authors concluded that “mild neurobiological dysfunctions in stress-related systems could help to explain diminished functionality and QoL in individuals with CM and MDD and contribute to the persistence of these impairments even after the remission of depressive symptoms”. The data are original and novel. I have a few minor concerns:

What is the use of presenting Models 1 to 4, in addition to the final model? I may have missed it but have not appreciated the benefit of presenting Models 1 to 4.

We included sequential MLR steps (models) with a hierarchical approach in order to verify changes in R2 (proportion of explained variance in the dependent variable in the model) when adding the selected variables to the equation in consecutive steps. We have clarified this issue in the Methods section.

Outliers have been excluded in Tables 2, 3 and 4. How are the outliers defined and how many outliers have been excluded? I think such information could be helpful to provide confidence in the solidity of the data.

As explained elsewhere [44], outliers were defined as transformed cortisol values that were located more than three standard deviations from the mean, and they were excluded from analyses. The number of outliers were 2 samples for the HC group and 7 samples for the non-remitted MDD group, without outliers in the remitted MDD group.

Line 180 to 182, “storage Samples were stored in refrigerators and returned personally by each participant. After the samples were received, they were stored at -20 °C and later sent to the BioBanc…” How soon did the participants return the samples once collecting them? How soon did the investigators send the samples out to the BioBanc once they receive them from participants? Variations of the storage durations of samples may affect the level of cortisol, so clarifying the time duration for these storage time would be useful.

Samples were returned within one week. The samples were stored at -20ºC the same day of receiving them. This information has been specified in the Methods.

Reviewer 2 Report

The submitted article by Salvat-Pujol et al. provides new, subtle evidence for the link between alterations in the HPA and persistence of impaired quality of life in MDD patients which have had experienced childhood maltreatment. The experiments have been well conceived and executed however this study overlaps significantly with their previous work (Hypothalamic-pituitary-adrenal axis activity and cognition in major depression: The role of remission status; doi.org/10.1016/j.psyneuen.2016.11.007). There are major and minor issues which need to be corrected before this manuscript can be accepted for publication.

Major issues:

I have a problem with methodology, as I understood; clinical, neuropsychological assessment and salivary cortisol measurement are the same for both manuscripts? If yes, this should be stated more clearly in the introduction and material and methods sections.

Minor issues:

Line 446: Cortisol measures…. Please use "Cortisol levels were measured… " instead.

Author Response

The submitted article by Salvat-Pujol et al. provides new, subtle evidence for the link between alterations in the HPA and persistence of impaired quality of life in MDD patients which have had experienced childhood maltreatment. The experiments have been well conceived and executed however this study overlaps significantly with their previous work (Hypothalamic-pituitary-adrenal axis activity and cognition in major depression: The role of remission status; doi.org/10.1016/j.psyneuen.2016.11.007). There are major and minor issues which need to be corrected before this manuscript can be accepted for publication.

Major issues:

I have a problem with methodology, as I understood; clinical, neuropsychological assessment and salivary cortisol measurement are the same for both manuscripts? If yes, this should be stated more clearly in the introduction and material and methods sections.

We have clarified this in the methods (lines 118-120): “Although the cognitive assessment and HPA axis measures of the sample are the same as in the previous study [44], the current manuscript tested different hypotheses and included new data on functionality and QoL.”

Minor issues:

Line 446: Cortisol measures…. Please use "Cortisol levels were measured… " instead.

Ok. We have changed the sentence as suggested.